# A Contemporary Review of the Use of Extracorporeal CytoSorb^®^ Hemoadsorption Therapy in Patients with Infective Endocarditis

**DOI:** 10.3390/jcm13030763

**Published:** 2024-01-29

**Authors:** Anan Gong, Yupei Li, Mei Yang, Shujing Wang, Baihai Su

**Affiliations:** 1Department of Nephrology, Kidney Research Institute, Frontiers Science Center for Disease-Related Molecular Network, West China Hospital, Sichuan University, Chengdu 610041, China; gonganan@stu.scu.edu.cn (A.G.); 2023224070050@stu.scu.edu.cn (S.W.); 2West China School of Medicine, Sichuan University, Chengdu 610041, China; 3General Practice Ward/International Medical Center Ward, General Practice Medical Center, West China Hospital, Sichuan University, Chengdu 610041, China; ym135688@163.com; 4Department of Nephrology, The First People’s Hospital of Shuangliu District, Chengdu 610200, China; 5Med+ Biomaterial Institute of West China Hospital/West China School of Medicine, Sichuan University, Chengdu 610041, China; 6Med-X Center for Materials, Sichuan University, Chengdu 610041, China

**Keywords:** hemoadsorption, infective endocarditis, CytoSorb^®^, cardiopulmonary bypass, cardiac surgery, mortality

## Abstract

Infective endocarditis (IE) is a rare but severe disease with high morbidity and mortality. Cardiac surgery plays a major role in the contemporary clinical management of IE patients. During cardiac surgery, cardiopulmonary bypass significantly contributes to an increased risk of organ dysfunction and mortality by inducing an acute inflammatory response, vascular endothelial cell injury, impairment of the coagulation cascade, and ischemia–reperfusion injury. During the past decade, the use of extracorporeal hemoadsorption therapy with the CytoSorb^®^ hemoadsorber (CytoSorbents Europe GmbH, Berlin, Germany) has been proposed as an adjuvant therapy to mediate inflammatory responses in IE patients undergoing cardiac surgery with cardiopulmonary bypass. However, there is currently no systematic evaluation of the effect of CytoSorb^®^ hemoadsorption on clinical outcomes such as hemodynamics, organ dysfunction, and mortality in patients with IE. Therefore, in this review, we exclusively discuss contemporary findings concerning the rationale, clinical evidence, and future perspectives for CytoSorb^®^ hemoadsorption therapy in IE patients.

## 1. Introduction

Infective endocarditis (IE), an infection of the endothelium of the heart, has an annual incidence of 15 cases per 100,000 population and carries a high 30-day mortality rate of up to 30% [1]. In patients with IE, bacteremia triggers complex interactions between microorganisms, platelets, diseased valvular endothelium, and host immunity, which contributes to vegetation and destruction of valvular or perivalvular tissue [2]. Accordingly, prolonged antibiotic therapy is mandatory for managing IE patients since valvular and perivalvular infections are difficult to control by host immunological responses and antibiotics [3]. Both the American Heart Association and European Society of Cardiology clinical guidelines have detailed the selection of an appropriate bactericidal treatment for IE patients in depth [4,5].

Beyond antibiotic therapy, cardiac surgery can restore normal valve function and resect infected tissues in IE patients with acute severe complications [2]. Approximately 50% of IE patients who develop severe complications, such as heart failure, severe valve dysfunction, prosthetic valve endocarditis, recurrent systemic embolization, large mobile vegetation, and persistent sepsis, require a cardiac operation [6]. During cardiac surgery, cardiopulmonary bypass (CPB) is established to temporarily replace the heart and lung functions of a patient [7]. Nevertheless, CPB is significantly associated with multiple pathological changes, including an acute inflammatory response, vascular endothelial cell injury, impairment of the coagulation cascade, and ischemia–reperfusion injury, which jointly contribute to multiple organ dysfunction and mortality in patients undergoing cardiac surgeries [8]. The acute inflammatory response is characterized by the release of proinflammatory cytokines, such as interleukin (IL)-1, IL-6, IL-8, and tumor necrosis factor-alpha (TNF-α), and is thought to play vital immunopathologic roles in CPB-associated complications [7,9]. Furthermore, a prospective case–control pilot study by Diab et al. demonstrated that patients with IE had higher inflammatory mediator levels than those without IE at the end of CPB and that the plasma level of IL-6 during CPB was significantly correlated with the severity of postoperative organ dysfunction in IE patients [10]. Therefore, it is reasonable to speculate that the removal of such circulating inflammatory mediators might help improve organ dysfunction in IE patients undergoing cardiac surgery with CPB.

Recently, the use of extracorporeal hemoadsorption therapy with the CytoSorb^®^ hemoadsorber has been proposed as an adjuvant therapy to mediate inflammatory responses by eliminating proinflammatory cytokines during cardiac surgery [9]. Although several studies have been conducted to evaluate the efficacy of CytoSorb^®^ (CytoSorbents Europe GmbH, Berlin, Germany) hemoadsorption in patients undergoing cardiac surgery, the results have been inconsistent [9,11,12,13,14,15,16]. For instance, two randomized controlled studies failed to demonstrate a decrease in either postoperative organ dysfunction or vasopressor use through CytoSorb^®^ hemoadsorption in IE patients [9,13]. In contrast, other retrospective studies found that intraoperative CytoSorb^®^ hemoadsorption significantly reduced sepsis-related mortality and improved hemodynamics and organ function [11,12]. Notably, there are several flaws in the study design, sample size, and CytoSorb^®^ prescriptions regarding initiation timing and treatment duration across these previous clinical studies, which unfortunately makes the interpretation of these results challenging. Considering that there is currently no systematic evaluation of the effect of CytoSorb^®^ on patients with IE, we are motivated to conduct a narrative review to exclusively discuss contemporary findings concerning the rationale, clinical evidence, and future perspectives for hemoadsorption therapy in IE patients undergoing cardiac surgery.

## 2. Rationale for Hemoadsorption Therapy in IE Patients Undergoing Cardiac Surgery

Cardiac surgery may activate the host immune system through surgical trauma, CPB, artificial surfaces, or ischemia–reperfusion injury, which might further lead to systemic inflammatory response syndrome (SIRS) [17,18]. In 2017, a retrospective cohort study enrolling 28,513 patients revealed that the overall prevalence of postoperative SIRS within the first 24 h after cardiac surgery was as high as 58.7% [19]. Another retrospective study also showed that 142 (28.3%) of 502 patients who underwent cardiac surgery with CPB fulfilled the SIRS criteria at 24 h and that SIRS was related to a more complicated postoperative course and greater postoperative morbidity [20]. IE patients who undergo CBP are also at a high risk of developing SIRS owing to the potential for intraoperative bacterial spread [14]. Therefore, it is reasonable to speculate that the control of unwanted SIRS might help to improve the outcomes of IE patients undergoing cardiac surgery with CPB.

Hemoadsorption refers to the circulation of blood through a hemoadsorber containing specific sorbents, with adsorption serving as the only mechanism for the removal of specific solutes or substances [21]. Hemoadsorption has been commonly used to remove inflammatory cytokines and metabolic wastes in multiple hyperinflammatory conditions (namely, sepsis, acute liver failure, acute pancreatitis, acute respiratory distress syndrome, severe COVID-19, etc.) or acute poisoning during the past two decades [22,23,24,25,26]. Currently, CytoSorb^®^ (CytoSorbents Europe GmbH, Berlin, Germany), HA 330 (Jafron Biomedical Co., Ltd., Zhuhai, China), and Toraymyxin™ (Toray Medical Co., Ltd., Tokyo, Japan) are three mainstream hemoadsorbers that are widely used in critical care settings [27]. Among them, CytoSorb^®^, a CE-approved hemoadsorber, can significantly remove cytokines, bilirubin, toxic bile acids, and myoglobin in the circulating blood [28,29,30,31]. Equipped with highly porous, biocompatible sorbent polystyrene divinylbenzene beads, CytoSorb^®^ cartridges have a total surface area of >45,000 m^2^ and are capable of adsorbing various hydrophobic cytokine molecules with a molecular weight ranging from 5 to 55 kDa [22].

In vitro studies indicate that CytoSorb^®^ can rapidly reduce the levels of multiple cytokines in experimental settings of endotoxemia [32,33]. However, the effect of CytoSorb^®^ hemoadsorption on hyperinflammation remains controversial across different clinical studies. Some observational studies suggest that CytoSorb^®^ may lower circulating cytokine concentrations, ameliorate organ dysfunction, and improve hemodynamics in critically ill patients with various hyperinflammatory conditions [34,35,36]. Similarly, a prospective, randomized single-center study by Garau et al. showed that a significant short-term decrease in the proinflammatory cytokine levels of IL-8 and TNF-α at 6 h after CPB could be observed in patients undergoing on-pump cardiac surgery and intraoperative CytoSorb^®^ hemoadsorption [37]. In contrast, another randomized controlled trial enrolling 30 patients undergoing elective cardiac surgery showed that CytoSorb^®^ hemoadsorption failed to reduce both pro- and anti-inflammatory cytokines [38]. More recently, Daniela et al. performed a retrospective study with 56 participants to investigate whether the use of CytoSorb^®^ has an effect on IL-6 levels in patients undergoing cardiac surgery [17]. The results showed that IL-6 levels peaked on the first postoperative day in both the CytoSorb^®^ and control groups (CytoSorb^®^: 775.3 ± 838.4 vs. control: 855.5 ± 1052.9 pg/mL) and that intraoperative CytoSorb^®^ hemoadsorption was not associated with a significant reduction in IL-6 levels or periprocedural mortality. A subgroup analysis of a recent meta-analysis also revealed that CytoSorb^®^ treatment did not lower mortality in patients who underwent cardiac surgery with CPB (0.91 [0.64; 1.29], RR [95%-CI]) [39]. To gather enough evidence for the safe use of CytoSorb^®^ in clinical practice, we are motivated to discuss contemporary clinical evidence for CytoSorb^®^ hemoadsorption therapy in IE patients undergoing cardiac surgeries in the Section 3.

## 3. Clinical Evidence for Hemoadsorption Therapy in IE Patients Undergoing Cardiac Surgery

As discussed above, CytoSorb^®^ hemoadsorption holds promise for attenuating inflammation during CPB sessions. Thus, several clinical studies have determined the efficacy of CytoSorb^®^ hemoadsorption in IE patients undergoing cardiac surgery, as shown in Table 1. In most scenarios, only a single CytoSorb^®^ cartridge was installed parallel to the venous CPB circuit for intraoperative hemoadsorption therapy, as shown in Figure 1. It should be noted that both baseline patient characteristics and hemoadsorption prescriptions varied significantly among these studies. For instance, blood flow may vary from 100 to 700 mL/min during extracorporeal hemoadsorption sessions, while the EuroScore II score, a well-established tool for evaluating cardiac operative risk, may range from 3.0 to 33.8 across these studies. These variations may contribute to the observed differences in the effect of CytoSorb^®^ hemoadsorption on both laboratory and clinical outcomes in IE patients. In this section, we mainly discuss the efficacy of CytoSorb^®^ hemoadsorption on patient-centered outcomes, such as mortality, organ dysfunction, hemodynamics, and inflammatory parameters.

The effect of intraoperative CytoSorb^®^ hemoadsorption on hemodynamics in IE patients undergoing cardiac surgery remains debatable. Early in 2017, Träger et al. first conducted a case series study to evaluate the efficacy of intraoperative CytoSorb^®^ treatment in 39 IE patients [40]. In this study, a single CytoSorb^®^ cartridge was integrated into the extracorporeal CPB circuit with a blood flow rate ranging from 200 to 400 mL/min. The median duration of CytoSorb^®^ hemoadsorption was 132 min. The results showed a remarkable increase in inflammatory mediators, including IL-6 and IL-8, after the surgical procedure. Intraoperative CytoSorb^®^ treatment was associated with a marked reduction in IL-6 and IL-8 plasma levels postoperatively and hemodynamic stability before, during, and after surgery compared with those in the historical group, as evidenced by a rapid decrease in the need for vasopressors. In patients receiving intraoperative CytoSorb^®^ treatment, the APACHE II score also decreased from a median of 31 postoperation to a median of 20 on day 1 postoperatively.

Recently, Haidari et al. included 130 patients with confirmed *S. aureus* IE to investigate the effect of intraoperative hemoadsorption on the vasoactive-inotropic score within the first 72 h after surgery [41]. In the hemoadsorption group, a CytoSorb^®^ cartridge was installed in a parallel circuit of the CPB machine, during which the blood flow rate ranged between 100 and 700 mL/min. The mean CPB time was 133.2 min in the hemoadsorption group and 142.4 min in the control group. Significantly decreased vasoactive-inotropic scores were observed in the hemoadsorption group vs. the control group at all time points. However, the difference in postoperative sequential organ failure assessment (SOFA) scores between the hemoadsorption group and the control group was not significant during the postoperative course.

In another small randomized, controlled, nonblinded clinical trial, Holmén et al. enrolled 19 IE patients who were undergoing valve surgery to determine the effect of CytoSorb^®^ hemoadsorption on hemodynamics [13]. In the CytoSorb^®^ group, one CytoSorb^®^ cartridge was integrated parallel to the standard CPB circuit with a median treatment duration of 137 min. CytoSorb^®^ hemoadsorption was related to an insignificantly reduced accumulated norepinephrine dose at postoperative time points compared to that in the control group (24 h: median 36 [25–75 percentiles; 12–57] μg vs. 114 [25–559] μg, *p* = 0.11; 48 h: 36 [12–99] μg vs. 261 [25–689] μg, *p* = 0.09). However, CytoSorb^®^ treatment significantly reduced the need for blood transfusions (285 [0–657] mL vs. 1940 [883–2148] mL, *p* = 0.03).

Furthermore, Asch et al. randomly assigned 20 IE patients to either the CytoSorb^®^ hemoadsorption group or the control group to investigate the effect of perioperative hemoadsorption therapy on inflammatory parameters and hemodynamics [14]. CytoSorb^®^ treatment was initiated intraoperatively and continued for 24 h postoperatively, with a median operation time of 264 min. Unfortunately, the authors failed to demonstrate a beneficial effect of CytoSorb^®^ treatment on hemodynamics in IE patients who underwent hemoadsorption intraoperatively and 24 h postoperatively [14]. Moreover, there were no significant differences in median cytokine levels (IL-6 or TNF-α) between the two groups during the perioperative course.

Currently, there are also several clinical trials evaluating the effect of CytoSorb^®^ on the survival and organ dysfunction of IE patients. In 2022, Kalisnik et al. included 202 high-risk patients with active left-sided IE to compare the incidence of postoperative sepsis, sepsis-associated death, and in-hospital mortality between CytoSorb^®^ and the standard of care [12]. The CytoSorb^®^ cartridge was installed into the venous system of the CPB between the oxygenator and venous reservoir for the entire duration of CPB. After propensity score matching, hemoadsorption significantly reduced the incidence of postoperative sepsis and sepsis-associated mortality compared to the standard of care (22.2% vs. 39.4%, *p* = 0.014 and 8.1% vs. 22.2%, *p* = 0.01, respectively). Patients in the hemoadsorption group tended to have lower in-hospital mortality than those in the control group, although the difference between the two groups was statistically insignificant (14.1% vs. 26.3%, *p* = 0.052). Multivariate regression analysis also confirmed that CytoSorb^®^ treatment was associated with decreased sepsis-associated (OR 0.09, 95% CI 0.013–0.62, *p* = 0.014) as well as in-hospital mortality (OR 0.069, 95% CI 0.006–0.795, *p* = 0.032). Furthermore, CytoSorb^®^ treatment was related to lower C-reactive protein levels 24 h after surgery, lower leukocyte counts on the second postoperative day, and lower blood transfusion requirements during the postoperative course.

In another retrospective single-center study, Santer et al. enrolled a total of 241 adult IE patients who had undergone cardiac surgery with CPB between January 2009 and December 2019 to evaluate the clinical benefits of CytoSorb^®^ therapy on in-hospital mortality and hemodynamics. A single CytoSorb^®^ cartridge was installed into the venous CPB tube at an average blood flow rate of 500 mL/min during the entire CPB duration. They found no significant difference in in-hospital mortality, major adverse cardiac or cerebrovascular events, or postoperative kidney failure between patients receiving hemoadsorption and those receiving standard of care [15]. More importantly, hemoadsorption was associated with an increased need for vasoactive agents, including norepinephrine (88.4 vs. 52.8%; *p* = 0.001) and milrinone (42.2 vs. 17.2%; *p* = 0.046). CytoSorb^®^ treatment also led to a higher incidence of reoperation for bleeding, as did increased postoperative demand for blood products (red blood cell concentrates and platelets) [15].

The REMOVE study, the largest multicenter, randomized, nonblinded, controlled trial thus far in this field, enrolled 288 IE patients undergoing cardiac surgery to further study the effect of hemoadsorption vs. standard of care on postoperative organ dysfunction as determined by the change in SOFA score [9]. The study also included 30-day mortality, duration of mechanical ventilation, and need for vasopressor and renal replacement therapy as secondary outcomes. For patients who were assigned to receive hemoadsorption, a CytoSorb^®^ cartridge was integrated into the venous line of the CPB circuit. The median CPB time was 128 min in the hemoadsorption group and 120 min in the control group. Finally, 282 patients (138 patients in the hemoadsorption group and 144 patients in the control group) were included in the modified intention-to-treat analysis. The total duration of hemoadsorption in the CytoSorb group was 2.31 ± 1.45 h. The results showed that hemoadsorption therapy failed to reduce organ dysfunction compared with standard of care (change in SOFA score: 1.79 ± 3.75 for the hemoadsorption group and 1.93 ± 3.53 for the control group; 95% CI, −1.30 to 0.83; *p* = 0.666), although the levels of IL-1β and IL-18 in the hemoadsorption group were significantly lower than those in the control group. Moreover, 30-day mortality did not differ between the hemoadsorption group and the control group (21% vs. 22%; *p* = 0.782). Hemoadsorption also did not reduce the duration of postoperative renal replacement therapy, mechanical ventilation, vasopressor therapy, or the length of ICU or hospital stay. Therefore, these results question a direct link between reducing plasma cytokine levels by hemoadsorption and improving patient-centered clinical outcomes, such as organ dysfunction and mortality.

In IE patients undergoing cardiac surgery, acute kidney injury is a common postoperative complication that might contribute to an increased risk of operative mortality [42]. Accordingly, Kühne studied whether IE patients who developed intraoperative acute kidney injury might benefit from additional postoperative CytoSorb treatment in a small case series [43]. In total, 20 patients who underwent CPB-assisted cardiac surgery for acute IE were assigned to either CytoSorb intraoperatively (Group 1) or CytoSorb intraoperatively plus postoperatively (Group 2), with a blood flow rate between 300 and 600 mL/min. At baseline, patients in Group 2 had more pronounced disease severity, as evidenced by a higher EuroSCORE II, a higher reoperation rate, more cardiopulmonary bypass times, and a worse inflammatory status than those in Group 1. Notably, the results showed that although additional postoperative CytoSorb treatment was associated with a higher rate of postoperative complications and a longer ICU stay, patients in the intraoperative plus postoperative hemoadsorption group had similar 90-day survival rates compared to those treated only intraoperatively, which suggested that postoperative continuation of CytoSorb hemoadsorption might be beneficial in IE patients who develop perioperative acute kidney injury.

Taken together, although the concept of cytokine elimination in IE patients undergoing cardiac surgery is tempting, there is no solid evidence to favor routine clinical application of intraoperative hemoadsorption in such patients. These conflicting results also remind us that we have the obligation to perform CytoSorb^®^ hemoadsorption in properly selected CPB patients, taking into account the treatment timing, duration, and dose. Whether longer durations or higher treatment doses of CytoSorb^®^ hemoadsorption may exert additional beneficial effects remains to be explored. It should also be noted that the majority of major clinical trials in this field included only European participants. Accordingly, external validation of the conclusions of the abovementioned clinical trials to guide the use of CytoSorb^®^ hemoadsorption worldwide is also needed in the future, taking into account differences in genetic information and clinical practice patterns across different populations and countries.

**Table 1 jcm-13-00763-t001:** Summary of major clinical trials evaluating the effect of CytoSorb^®^ hemoadsorption in patients undergoing cardiac surgery with CPB.

Author, Publication Year	Study Location	Study Design	Study Period	Sample Size	Mean or Median EuroScore II	Hemoadsorption Prescription	Main Findings
Silke Asch, 2021 [14]	Göttingen, Germany	RCT	November 2018 to March 2020	20	Cytosorb: 8.5 Control: 3.6	Cytosorb^®^ hemoadsorption was initiated intraoperatively and continued for 24 h postoperatively.	Cytosorb^®^ hemoadsorption neither resulted in a reduction of inflammatory parameters nor an improvement of hemodynamics in IE patients.
Mahmoud Diab, 2022 [9]	Multicenter, Germany	RCT	17 January 2018 to 31 January 2020	282	Cytosorb: 19.1 ± 17.3 Control: 20.2 ± 17.8	Hemadsorption during CPB.	Although Cytosorb^®^ hemoadsorption reduced plasma cytokines, there was no difference in clinically relevant outcome measures and no reduction in postoperative organ dysfunction.
Ingo Garau, 2019 [37]	Hamburg, Germany	RCT	September 2013 to June 2015	40	Cytosorb: 6.1 Control: 6.3	Hemadsorption during CPB with a blood flow of 300 mL/min.	In elective on-pump cardiac surgery patients, Cytosorb^®^ hemoadsorption was associated with a short-term reduction in pro-inflammatory cytokine levels of IL-8 and TNFα.
Elettra C Poli, 2019 [38]	Lausanne, Switzerland	RCT	May 2016 to January 2018	30	Cytosorb: 3.0 Control: 5.1	Hemadsorption during CPB.	CytoSorb^®^ hemoadsorption during CPB was not associated with a decrease in pro- or anti-inflammatory cytokines nor with an improvement in relevant clinical outcomes.
Anna Holmen, 2022 [13]	Gothenburg, Sweden	RCT	April 2019 to September 2020	19	NA	Hemadsorption during CPB.	Cytosorb^®^ hemoadsorption contributed to an insignificantly decreased vasopressor use after surgery in IE patients.
Zaki Haidari, 2023 [41]	Essen and Nuremberg, Germany	Retrospective study	January 2015 to March 2022	130	Cytosorb: 11.9 Control: 12.0	Hemadsorption during CPB with a blood flow ranging from 100 to 700 mL/min.	Intraoperative Cytosorb^®^ hemoadsorption significantly contributed to reduced sepsis-associated mortality, 30- and 90-day mortality, and improved hemodynamics in high-risk IE patients.
Jurij Matija Kalisnik, 2022 [12]	Nuremberg, Germany	Retrospective study	January 2015 to April 2021	202	Cytosorb: 9.89 Control: 8.95	Hemadsorption during CPB.	After propensity score match, intraoperative Cytosorb^®^ hemoadsorption significantly reduced sepsis and sepsis-associated mortality after cardiac surgery in high-risk patients with active left-sided native and prosthetic valve IE.
David Santer, 2021 [15]	Basel, Switzerland	Retrospective study	2009 to 2019	241	Cytosorb: 7.8 Control: 8.6	Hemadsorption during CPB with a blood flow of 500 mL/min.	Intraoperative Cytosorb^®^ hemoadsorption did not reduce in-hospital mortality, incidence of delirium, myocardial ischemia, stroke, and postoperative renal failure, but was significantly associated with increased in-hospital stay and incidence of reoperation for bleeding in IE patients.
Lars-Uwe Kühne, 2019 [43]	Hamburg, Germany	Case series	July 2017 to April 2018	20	Group 1: 26.8 Group 2: 33.8	Group 1: intraoperative hemoadsorption with a blood flow rate between 300 and 600 mL/min. Group 2: intraoperative plus postoperative hemoadsorption with a blood flow rate between 300 and 600 mL/min.	IE patients undergoing intraoperative plus postoperative Cytosorb^®^ hemoadsorption showed a similar ICU and 90-day survival compared to those undergoing intraoperative Cytosorb^®^ hemoadsorption only. However, postoperative continuation of hemoadsorption treatment was associated with a higher rate of postoperative complications and a longer intensive care unit stay despite a significant difference in baseline disease severity between the two groups.
Karl Träger, 2017 [40]	Ulm, Germany	Case series with a historical group	September 2013 to August 2016	67	Cytosorb: 11 Historical control: 9 for ICU survivors	Hemadsorption during CPB with a blood flow ranging from 200 to 400 mL/min.	Intraoperative Cytosorb^®^ hemoadsorption contributed to reduced plasma IL-6 and IL-8 levels and improved hemodynamics in IE patients.

Abbreviations: IE, infective endocarditis; CPB: cardiopulmonary bypass; ICU: intensive care unit; IL: interleukin.

## 4. Safety Concerns

Generally, CytoSorb^®^ hemoadsorption was safe and well tolerated, with no device-related adverse events during or after CPB sessions [12,38,40]. Major safety concerns associated with the use of CytoSorb^®^ in clinical practice include the nonspecific adsorption of antibiotics, anticoagulants, and coagulation factors [38,44,45,46]. Specifically, several small observational studies have shown that CytoSorb^®^ treatment significantly reduces vancomycin levels in critically ill patients [45,47]. Considering the important role of antibiotics in the management of IE, dose adjustment of antibiotics should be considered in IE patients undergoing CPB with CytoSorb^®^ hemoadsorption, especially in those with prolonged postoperative CytoSorb^®^ hemoadsorption. Interestingly, despite several case reports highlighting a promising approach to reduce bleeding risk during cardiac surgery by intraoperative removal of ticagrelor or rivaroxaban [48,49], Santer et al. argued that CytoSorb^®^ hemoadsorption might contribute to increased bleeding risk and the need for blood transfusion through its nonspecific adsorption of coagulation factors. Therefore, the safety of CytoSorb^®^ for use in IE patients undergoing cardiac surgery and CPB should be further evaluated in well-designed large randomized controlled trials.

## 5. Health Economics

Intraoperative hemoadsorption during CPB in IE patients might have economic benefits due to a reduced length of ICU stay, which might further lead to improved healthcare resource use [50]. Using data from the German healthcare system, Cristina et al. developed an Excel-based budget impact model to simulate the patient course over the ICU stay in IE patients. In the base-case scenario, CytoSorb^®^ hemoadsorption resulted in a savings of EUR 2298 per patient. In the case of full device-specific reimbursement, the savings could increase to EUR 3804 per patient. Furthermore, the deterministic and probabilistic sensitivity analyses confirmed the robustness of savings. The study has several limitations. First, the cost associated with antibiotic therapy adjustment during CytoSorb^®^ hemoadsorption and the length of in-hospital stay, which might also have an impact on the final calculated savings, were not taken into consideration. Second, recent high-quality studies have not shown a beneficial effect of CytoSorb^®^ treatment on shortening the length of ICU stay, which is crucial for developing a budget impact model. Therefore, these findings must be confirmed by further prospective analyses reporting definite benefits in terms of reduced ICU stays.

## 6. Conclusions

Infective endocarditis (IE) is a rare but severe disease with high mortality and healthcare burdens. Cardiac surgery plays a vital role in the clinical management of IE patients; however, it also contributes to the unexpected activation of inflammatory responses. Hemoadsorption with the CytoSorb^®^ hemofilter theoretically holds promise as an adjuvant treatment by attenuating inflammation during CPB sessions. Unfortunately, the effect of CytoSorb^®^ hemoadsorption on patient-centered outcomes such as hemodynamics, organ dysfunction, length of ICU stay, and in-hospital mortality remains controversial in IE patients who undergo cardiac surgery with CPB, which prevents the routine clinical use of CytoSorb^®^ hemoadsorption in this unique patient population. Therefore, additional evidence from large, well-designed randomized controlled trials, including the timing of initiation, treatment duration, frequency of filter change, and dose adjustment of antibiotics, is urgently needed to clarify the best patient group to benefit from CytoSorb^®^ hemoadsorption in the future.

## Figures and Tables

**Figure 1 jcm-13-00763-f001:**
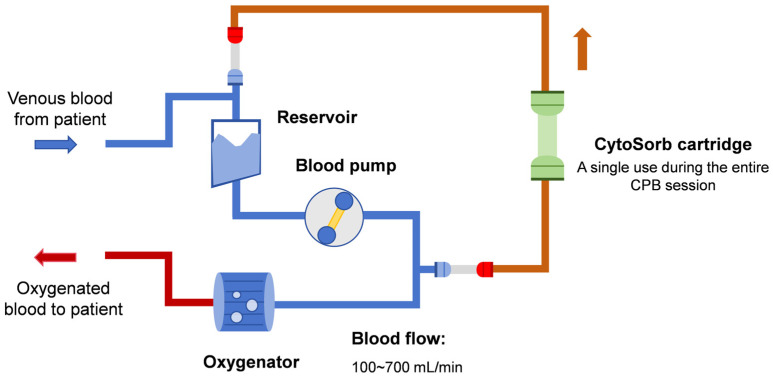
Scheme of integrating CytoSorb^®^ to a cardiopulmonary bypass circuit.

## Data Availability

No new data were created or analyzed in this study. Data sharing is not applicable to this article.

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
