# Peer review of "A Contemporary Review of the Use of Extracorporeal CytoSorb® Hemoadsorption Therapy in Patients with Infective Endocarditis"

_jcm, 2024, doi:10.3390/jcm13030763_

Round 1
Reviewer 1 Report
Comments and Suggestions for Authors
well thought review article.
One major flaw is the fact that when citing a paper one should use the surname of the first author and not his first name.
As the article is about cytosorb I think that there is no need for the paragraph (196-213) as it only confuses
abstract
line 16 "high morbidity and mortality"
line 23 systematic not systemic
line 40 "appropriate treatment"
line 41 is misleading one can deduce that all patients with IE will need surgery
line 70 non elegible heterogeneities should be replaced with flaws
line 79 delete"of"
Comments on the Quality of English Languagepatients do nor receive surgery they undergo surgery
Author Response
Comment 1: Well thought review article.
Response: Thank you for your valuable scientific comments and suggestions for improving the quality of our manuscript. Please find our detailed point-to-point responses to your comments below. Thank you very much for reviewing our manuscript. We look forward to receiving a positive response.
Comment 2: One major flaw is the fact that when citing a paper one should use the surname of the first author and not his first name.
Response: Thank you for your valuable suggestion. We have changed the citation style throughout the manuscript according to your suggestion.
Comment 3: As the article is about cytosorb I think that there is no need for the paragraph (196-213) as it only confuses.
Response: Thank you for your valuable scientific comment. The discussion on the use of HA330 in IE patients has been removed from the revised manuscript to avoid any confusing information.
Comment 4: line 16 "high morbidity and mortality"
Response: Thank you for your valuable scientific comment. The related sentence has now been changed to “Infective endocarditis (IE) is a rare but severe disease with high morbidity and mortality”.
Comment 5: line 23 systematic not systemic
Response: Thank you for your valuable scientific comment. We have corrected this mistake as per your suggestion. The sentence now reads “there is currently no systematic evaluation of the effect of CytoSorb® hemoadsorption on clinical outcomes...”
Comment 6: line 40 "appropriate treatment"
Response: Thank you for your valuable scientific comment. We have changed our statement according to your suggestion. The related sentence reads “Both the American Heart Association and European Society of Cardiology clinical guidelines have detailed the selection of an appropriate bactericidal treatment for IE patients in depth”.
Comment 7: line 41 is misleading one can deduce that all patients with IE will need surgery
Response: Thank you for your valuable suggestion. We have rephrased our sentence to avoid any misleading information. The text now reads “Beyond antibiotic therapy, cardiac surgery can restore normal valve function and resect infected tissues in IE patients with acute severe complications”.
Comment 8: line 70 non elegible heterogeneities should be replaced with flaws
Response: Thank you for your valuable suggestion. We have rephrased our sentence according to your suggestion. The text now reads “Of note, there are several flaws in the study design, sample size, and CytoSorb® prescriptions...”.
Comment 9: line 79 delete"of"
Response: Thank you for your valuable suggestion. This mistake has been corrected.
Comment 10: patients do nor receive surgery they undergo surgery
Response: Thank you for your valuable suggestion. We have changed our statements throughout the manuscript.
Reviewer 2 Report
Comments and Suggestions for Authors
The author presented a review on the use of Cytosorb for hemoadsorption in patients with Infective Endocarditis (IE). This is a relatively new concept, and the papers reviewed by the author were mostly published in recent years. I have the following suggestions:
First, a review article should not only present the results of various studies but also discuss the reasons behind them. It should address factors such as the different Cytosorb flow rates, durations, and inclusion criteria used in each study and explore whether these variations may contribute to the observed differences. Although this information is included in the tables, it is not explicitly discussed in the text, which is a missed opportunity.
Second, when discussing the various studies, the review article should be organized into sections that outline how each study used Cytosorb, the results of Cytosorb versus the control group, including both laboratory and clinical outcomes. Furthermore, it should delve into the safety of Cytosorb and conclude with an examination of its economic benefits. Separating these major topics will help readers grasp the key points, rather than presenting all study results in a single paragraph.
Third, as a review article, the current text appears to be insufficient in length. I believe that if the author incorporates the above two suggestions and makes substantial revisions, the article will likely reach an appropriate length.
Author Response
Comment 1: The author presented a review on the use of Cytosorb for hemoadsorption in patients with Infective Endocarditis (IE). This is a relatively new concept, and the papers reviewed by the author were mostly published in recent years. I have the following suggestions.
Response: Thank you for your valuable scientific comments and suggestions for improving the quality of our manuscript. Please find our detailed point-to-point responses to your comments below. Thank you very much for reviewing our manuscript. We look forward to receiving a positive response.
Comment 2: First, a review article should not only present the results of various studies but also discuss the reasons behind them. It should address factors such as the different Cytosorb flow rates, durations, and inclusion criteria used in each study and explore whether these variations may contribute to the observed differences. Although this information is included in the tables, it is not explicitly discussed in the text, which is a missed opportunity.
Response: Thank you for your valuable scientific comments. We agree with your point that different CytoSorb flow rates, durations, and inclusion criteria may contribute to the observed differences in the effect of CytoSorb® hemoadsorption on both laboratory and clinical outcomes in IE patients. We have added more discussion on the differences in CytoSorb prescriptions across the included studies. In most studies, only a single CytoSorb® cartridge was installed parallel to the venous CPB circuit for intraoperative hemoadsorption therapy, and the mean or median CPB time was approximately 2 hours. Therefore, we have added a figure to further show the readers the scheme and key parameters for combining CytoSorb® hemoadsorption with CPB sessions. Please see the related changes from Line 129 to 263 of the revised manuscript, which are highlighted in red for your convenience.
Comment 3: Second, when discussing the various studies, the review article should be organized into sections that outline how each study used Cytosorb, the results of Cytosorb versus the control group, including both laboratory and clinical outcomes. Furthermore, it should delve into the safety of Cytosorb and conclude with an examination of its economic benefits. Separating these major topics will help readers grasp the key points, rather than presenting all study results in a single paragraph.
Response: Thank you for your valuable scientific comments. We have reorganized our main text and added more discussion on the CytoSorb prescription used in each study and the effect of CytoSorb hemoadsorption on both laboratory and clinical outcomes. Furthermore, we have divided the discussion on the safety and potential economic benefits of CytoSorb into two separate parts according to your kind suggestion. Please see the related changes from Line 129 to 263 of the revised manuscript. We believe that these substantial revisions will help improve the readability of our paper.
Comment 4: Third, as a review article, the current text appears to be insufficient in length. I believe that if the author incorporates the above two suggestions and makes substantial revisions, the article will likely reach an appropriate length.
Response: Thank you for your valuable scientific comments. We have made substantial revisions to the main text of our manuscript according to your kind suggestions. The word count of the revised main text now exceeds 4000, which may help to make our revised manuscript more comprehensive and informative. of the revised main text now exceeds 4000, which helps to make our study more comprehensive. Thank you very much for reviewing our manuscript again. We look forward to receiving a positive response from you.
Reviewer 3 Report
Comments and Suggestions for Authors
The Authors provided an interesting narrative review of the use of extracorporeal CytoSorb hemoadsorption therapy in patients with infective endocarditis.
The paper is well written and can be clearly read throughout the text. As correctly stated by the Authors, the heterogeneity of indications, strategies, and patients' characteristics utilized by the studies investigating this topic, greatly affects the comparison between them to yield a unified conclusion.
The manuscript could benefit from the following comments:
Introduction
- The ESC guidelines have just released the 2023 IE update. I suggest changing reference 5 to this updated version.
Language
- When referring to a paper by first Author, I suggest using the surname, not the name. Correction to lines 111, 139, 146, 154, 159, 170, 171, 200, 228 is warranted.
- Correction of the “ROMOVE” to “REMOVE” study should be made on line 178.
Author Response
Comment 1: The Authors provided an interesting narrative review of the use of extracorporeal CytoSorb hemoadsorption therapy in patients with infective endocarditis. The paper is well written and can be clearly read throughout the text. As correctly stated by the Authors, the heterogeneity of indications, strategies, and patients' characteristics utilized by the studies investigating this topic, greatly affects the comparison between them to yield a unified conclusion. The manuscript could benefit from the following comments.
Response: Thank you for your valuable scientific comments and suggestions for improving the quality of our manuscript. Please find our detailed point-to-point responses to your comments below. Thank you very much for reviewing our manuscript. We look forward to receiving a positive response.
Comment 2: Introduction- The ESC guidelines have just released the 2023 IE update. I suggest changing reference 5 to this updated version.
Response: Thank you for your valuable suggestion. We have updated our citation as per your suggestion.
Comment 3: Language- When referring to a paper by first Author, I suggest using the surname, not the name. Correction to lines 111, 139, 146, 154, 159, 170, 171, 200, and 228 is warranted.
Response: Thank you for your valuable suggestion. We have changed the citation style throughout the manuscript according to your suggestion.
Comment 4: Correction of the “ROMOVE” to “REMOVE” study should be made on line 178.
Response: Thank you for your valuable suggestion. We have corrected this mistake in the revised manuscript.
Round 2
Reviewer 2 Report
Comments and Suggestions for Authors
The author made modifications according to the suggestions, and I have no other comments.